# communications
## earth & environment

# Projected decrease in trail access in the Arctic

J. D. Ford [1✉], D. G. Clark [2], L. Copland [3], T. Pearce[4], IHACC Research Team* & S. L. Harper [5]

Transportation systems in northern Canada are highly sensitive to climate change. We project how access to semi-permanent trails on land, water, and sea ice might change this century in Inuit Nunangat (the Inuit homeland in northern Canada), using CMIP6 projections coupled with trail access models developed with community members. Overall trail access is projected to diminish, with large declines in access for sea ice trails which play a central role for Inuit livelihoods and culture; limits to adaptation in southern regions of Inuit Nunangat within the next 40 years; a lengthening of the period when no trails are accessible; and an unequal distribution of impacts according to the knowledge, skills, equipment, and risk tolerance of trail users. There are opportunities for adaptation through efforts to develop skillsets and confidence in travelling in more marginal environmental conditions, which can considerably extend the envelope of days when trails are accessible and months when this is possible. Such actions could reduce impacts across emissions scenarios but their potential effectiveness declines at higher levels of global warming, and in southern regions only delays when sea ice trails become unusable.

[1] Priestley International Centre for Climate, University of Leeds, Leeds, UK. [2] Canadian Climate Institute, Vancouver, BC, Canada. [3] Department of Geography, Environment and Geomatics, University of Ottawa, Ottawa, ON, Canada. [4] Department Geography, Earth, and Environmental Sciences, University of Northern British Columbia, Prince George, BC, Canada. [5] School of Public Health, University of Alberta, Edmonton, AB, Canada. *A list of authors and their affiliations appears at the end of the paper. ✉email: j.ford2@leeds.ac.uk

For Indigenous Peoples and local communities across the Arctic, rapid changes in climate have affected the ability to safely use semi-permanent trails on sea ice, lakes, rivers, ocean, and tundra (herein referred to as "trails")[1]. These trails are often the only way of traveling from settlements to areas where people engage in hunting, trapping, fishing, and herding activities[2]. The very use of trails plays a foundational role in wellbeing and culture for many Indigenous Peoples, as well as underpinning the development and the exchange of Indigenous knowledge, with movement over such routes fundamental in how environmental change is confronted and experienced[3–5]. Routes taken define the circumstances through which animals are encountered and provide a connection to the past, often following archaeological traces (Bravo[6], Walls, Hvidberg et al.[7], Ward, Hill et al.[8]).

Documented impacts of climate change on trails include a reduction in the number days when trails are usable; increasing rates of search and rescue due to more dangerous conditions; a reduced ability to harvest traditional foods with associated nutritional security implications; a loss of access to sites associated with healing and self-healing practices, with wide-ranging implications for culture, health, and well-being[9–13]. These impacts have been compounded by changes in how people travel and different levels of knowledge among some trail users, particularly that which pertains to knowledge and use of the environment[14,15]. Future warming has the potential to magnify the risks of using trails, although the dynamics are not well understood, with no studies projecting or quantifying potential future impacts of climate change on trail use[3].

In this paper we project how future global warming might affect the ability to use trails, focusing on regions that make up Inuit Nunangat (homeland) of northern Canada. Inuit Nunangat makes up 35% of the Canadian landmass, with travel by all-terrain vehicle (ATV), small watercraft, snowmobile, and foot common across the Nunangat's 53 communities (Fig. 1). This involves the use of extensive networks of trails on land, water, or sea ice and often involves travelling hundreds of kilometres in remote regions[16–18]. Trail use is influenced by a diversity of climatic and non-climatic factors, including sea ice thickness and stability, wind conditions, precipitation, visibility, knowledge, and skillsets of trail users, financial resources, and access to technology[19–21]. Together these factors affect whether travel is possible and safe—what we term 'trail access.' To reflect the complex interplay between climate and sociocultural factors, we use a 7-step ethnoclimatology modeling framework that connects the knowledge of trail users and CMIP6 global climate simulations to project how trail access might be affected at different levels of warming (see Methods). We began our analysis by creating trail access models which specify quantitative real-world thresholds of weather and sea ice conditions for safe trail access, developed from semi-structured interviews and validation with Inuit trail users in communities across Inuit Nunangat, as reported in Ford et al.[22]. Unique models were created specifically for land, water, and sea ice trails. We also used different sets of thresholds (termed trail 'user type') to capture non-climatic factors affecting trail access, such as knowledge, skills, equipment, and risk tolerance of trail users, based on users' judgements about their competence and risk aversion in relation to ice conditions, temperature, precipitation, wind, and visibility. We group these characteristics into three categories which we label as: average risk tolerance (Type 1), low-risk tolerance (Type 2), and high-risk tolerance (Type 3). These archetypes are not intended to capture all users, but to capture a broad continuum of highly nuanced sets of individual characteristics (S1).

Next, we used gridded CMIP6 global climate model outputs to examine on how many days the trail access thresholds are exceeded between 2015–2100 (31,390 days) for regions around all 53 communities, using daily gridded climate projections of mean temperature, precipitation, mean surface wind, sea ice concentration, and sea ice thickness (see Methods). Data from five global climate models (GCMs) and for both a low (SSP245) and a high emissions scenario (SSP585) were used. We then ran each distinct GCM and emissions scenario combination (twelve in total given the five GCMs, one ensemble, and two emissions scenarios) through the nine trail access models (Land 1, 2, 3; Water 1, 2, 3; Ice 1, 2, 3).

## Results

**Projected changes in climate and sea ice conditions: 2015–2100.** Significant warming is projected across Inuit Nunangat by the end of the century, ranging from +4.8 °C to +8.6 °C compared to now (2015-2030 model average) (Fig. 2a). Both emissions scenarios have a consistent increase up until mid-century, after which mid-century warming under SSP245 levels off while warming continues under SSP585. The number of cold extremes (days below −30 °C) is projected to decrease from an average of about 20 days/year between 2015 and 2030 to about 2 days/year by mid-century. In SSP585 most regions do not have any days below −30 °C by end of the century. Temperature increase is less pronounced in more southerly regions of Inuit Nunangat (see Fig. S1, supplementary materials).

Wind and precipitation patterns are projected to shift. Models indicate there will generally be more extreme wind days when wind reaches >25 km/hr by the end of the century, particularly under SSP585 (Fig. 2b). The greatest change in average wind speed is projected at higher latitudes (Baffin, Ellesmere, and Inuvialuit Settlement Region), while some models show that average wind speeds may decrease slightly in Nunatsiavut. Average annual precipitation is projected to increase across all regions of Inuit Nunangat. The largest percent change is projected under SSP585 where our analysis shows an increase of 28.1% by end of century compared to the 2015-2030 baseline. Under SSP245 average precipitation increases by 16.6% by end of the century. Precipitation changes are even more dramatic for the higher latitude regions (see Fig. S2, supplementary materials).

Sea ice thickness and concentrations are quickly decreasing across Inuit Nunangat. Focusing on near-shore areas that are pertinent to trail users (within 150 km of communities), projections indicate that the annual average sea ice concentration will decline from about 50% of coverage to about 40% coverage by mid-century under both emissions scenarios (Fig. 2c). Models indicate that by 2100, annual average sea ice concentration across Inuit Nunangat will be less than 10% (see supplementary materials for trends by region). This trend signifies both shorter sea ice seasons and an expanding area at lower latitudes that will not have consistent ice coverage at all (i.e. Nunavik and Nunatsiavut). Similarly, we observe that models project that sea ice thickness will decrease rapidly. Under SSP585, average ice thickness in coastal waters of Nunatsiavut are projected to be <35 cm for about 220 days/year at present day, which is projected to increase to 283 days/year by end of the century under SSP245 and to every day of the year under SSP585 (Fig. 2d) (Fig. S2, supplementary materials).

**Projected trail access changes: 2015–2100.** Our models show that the effects of climate change on trail access varies widely among the categories of user and across regions, emissions scenario, and trail modes. We begin by focusing on the aggregate results for Inuit Nunangat using the ensemble model (mean values of all five GCMs) for the average trail users (Type 1), and then describe trends observed for each of these modelling

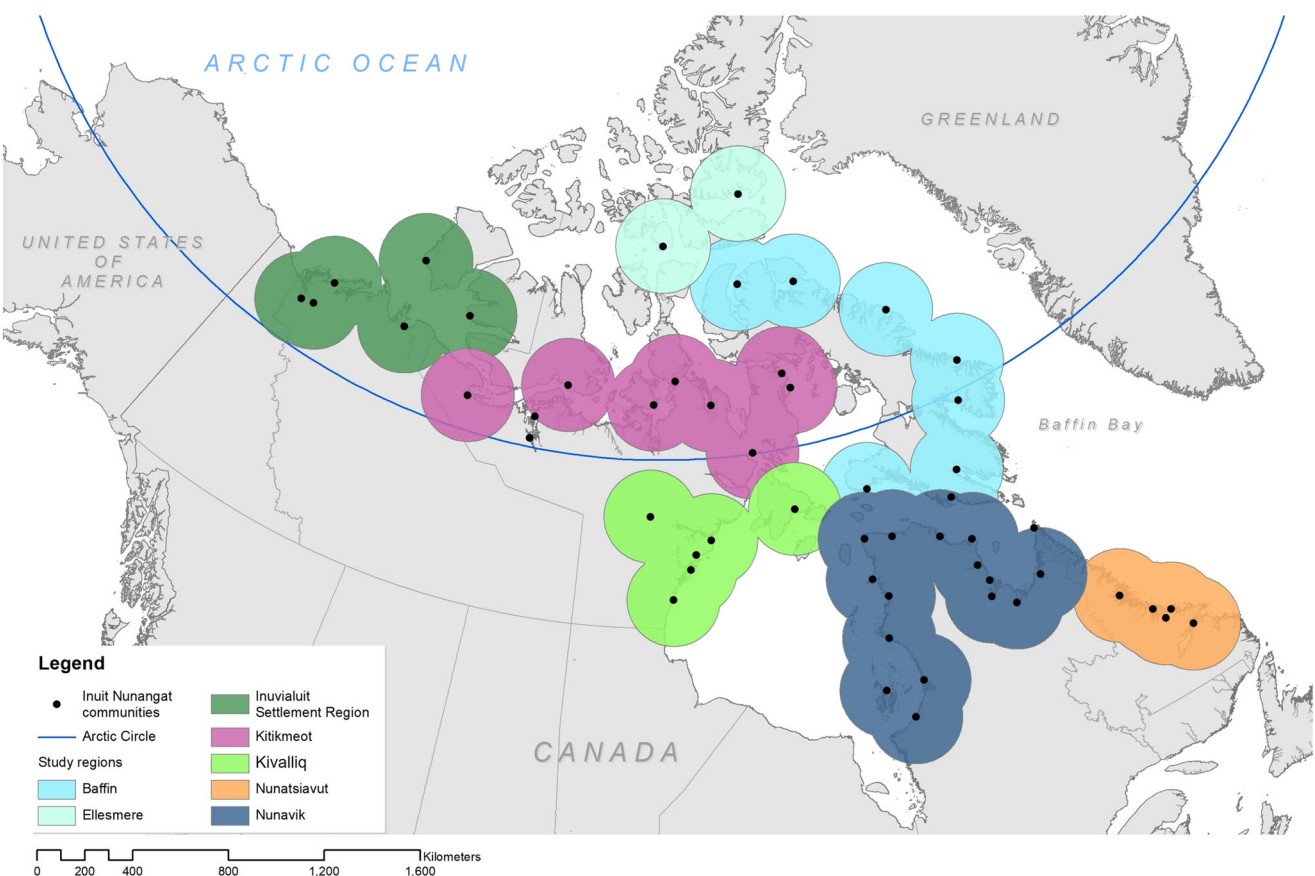

**Fig. 1 Inuit Nunangat (the Inuit homeland).** The region is home to approximately 56,000 people, inhabiting 53 mostly coastal communities. We examine trail access trends across 7 geographic regions, aggregating from analyses focusing on projected climatic and ice trends within a 100 km (climate) to 150 km (ice) radius of each community. The 150 km radius ice areas are shown here.

parameters below. Differences according to climate model (GCM) and additional results are provided in supplementary materials. Given that the modeling framework is based on general associations between regional climate-related conditions and trail access we only provide analyses for the primary geographical regions shown in Fig. 1, and do not disaggregate to the community-level, focusing on the key changes projected.

*Aggregate trends.* Across trail modes, our models project that the number of days each year where at least one mode of trail use is possible could shrink by 17 to 40 days/year (−7% and −16%) for the average trail user (Type 1) by 2100, depending on the emissions scenario (Fig. 3). Currently our models indicate an average of 249 days/year when at least one trail mode is usable (average for 2015 to 2030). Under SSP245 the number of good days decreases to about 236 days/year by mid-century and then remains fairly stable. Under SSP585, the number of good days continues to decline past 2050 to about 209 good days/year by the end of century.

*Trends by trail type.* Trail access shifts depending on the trail type (Fig. 4). Our models show that access to water trails will increase for all trail users and in all regions by the end of the century. On average, the number of good water trail days is projected to increase from about 86 days/year to 117 days/year under SSP245 and 148 days/year under SSP585 (+36% to +72%). The increase in water trail access primarily comes from better conditions in June and October, and under SSP585 water access also opens up in May and November. As a ratio, our models suggest that ice

conditions become less of a constraint over time for water trails, and wind speed becomes more of a constraint. Little overall change is projected in access to land trails in either scenario, with temperature remaining the leading cause of land trails not being accessible, followed by wind conditions.

We project a substantial decline in sea ice trail access in all regions across Inuit Nunangat. Our models show that good ice trail days will decline from an average of about 146 days/year currently to 105 days/year under SSP245, and 61 days/year under SSP585 (−28% to −58%) by end of century. The decline in sea ice trail access is most prominent in October, November, and May, and under SSP585 sea ice trails will also become less dependable in December and April by the end of the century.

*Regional differences.* Climate change impacts on trail access across Inuit Nunangat show large differences in both current trail access and future shifts based on latitude. We project that Type 1 users currently have 319 days/year where at least one mode of trail is available in Baffin, while the average in Nunatsiavut is 208 days/year. We project that the number of days with at least one good mode of trail accessible will decrease to 314 days/year and 312 days/year under SSP245 and 585 respectively for Baffin by 2100, while increasing to 216 days/year and 220 days/year for Nunatsiavut over the same period.

Sea ice trail conditions, in particular, are highly influenced by latitude. For example, at the southern edge of the Inuit Nunangat under SSP585, Nunatsiavut is projected to essentially have no sea ice trail access after 2060, with SSP245 extending access by about a decade. The knowledge, skills, equipment, and risk tolerance of

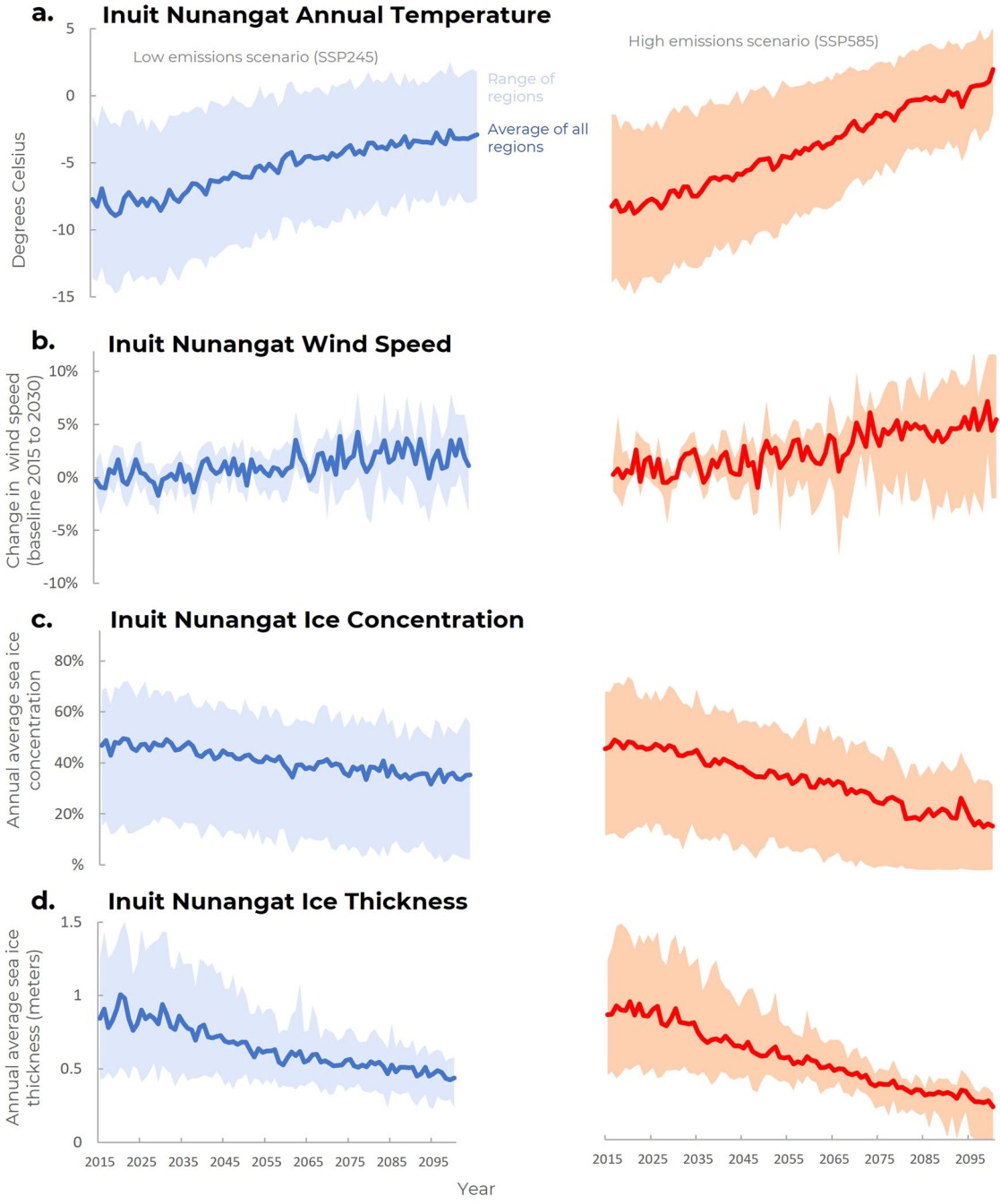

**Fig. 2 Projections of changes in climate (100 km radius) and sea ice (150 km radius) conditions around Inuit Nunangat's 53 communities.** Projections use the ensemble mean of the five study GCMs for SSP245 (left) and SSP585 (right), and are for year 2015 to 2100. The area range captures the variability across communities in Inuit Nunangat, while the solid line reflects the mean for all areas around communities. **a** average annual temperature; **b** percent change in average annual windspeed based on the 2015–2030 average; **c** annual average sea ice concentration; **d** annual average sea ice thickness.

trail users are unable to offset these declines, with Type 3 users only having access a few years longer than the average user. Further, we project that the lower skill level users or more risk averse (Type 2) will have no access in most years by mid-century under both low and high emissions scenarios. At the northern extremes (i.e., Baffin and Ellesmere), even under SSP585 Type 2 users maintain some access to sea ice trails by the end of the century, albeit at a lower level than present, while under SSP245 Type 1 and 3 users maintain >150 days/year of access by 2100 (approx. 80 days/year and 100 days/year SSP585 for Type 1 and 3 users, respectively). Results for land and water trails by region are presented in supplementary materials.

*User type.* User type has substantial influence on trail accessibility, with the average number of days each year where at least one trail mode is currently usable ranging from 41 days/year for Type 2 users (lower skill set, poorer equipment, and/or more risk averse) to 365 days/year for the Type 3 users (high skill levels, good equipment, and/or less risk averse). We project that the average user currently has 249 days/year where at least one trail is usable, with the average change in total trail access (where there is at least one usable trail) decreasing for Type 1 users in both the low (a loss of 17 days/year) and high emissions scenarios (a loss of 40 days/year) by 2090–2100 compared to the 2015–30 average. However, there are projected gains for the Type 2 users, with

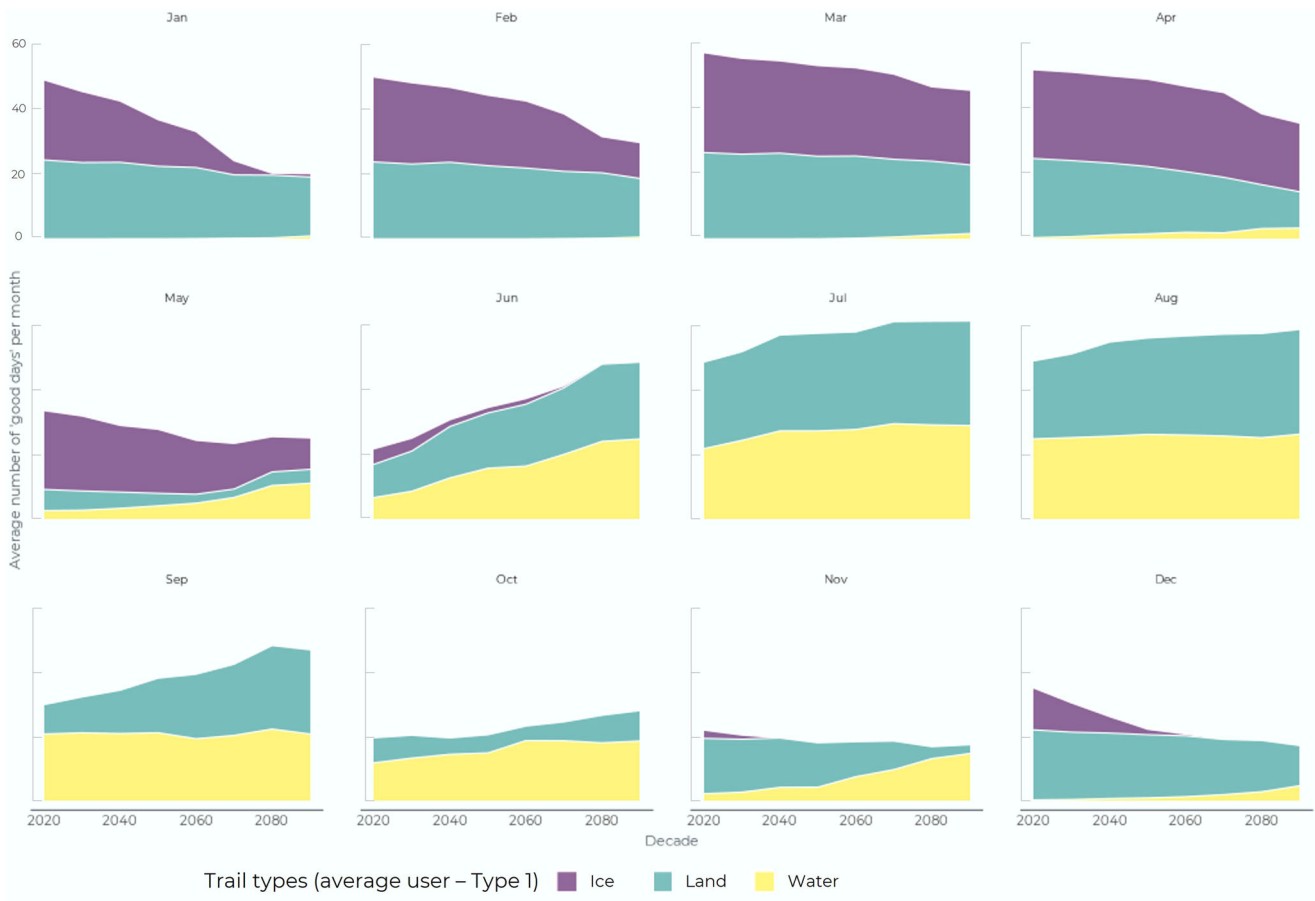

**Fig. 3 Monthly number of 'good days' when a trail is usable.** This is modeled for Type 1 (average) trail users for all trail types for SSP585 and averaged across all communities (decadal averages; e.g. "2020" is year 2020 to 2029).

models showing that they could gain an average of 16 days/year of trail access under SSP245 and 41 days/year under SSP585 between the 2015-2030 average and end of century. Because our model indicates that Type 3 users already have 365 days of access each year, there is no change projected (noting that this does not necessarily mean such users can get everywhere all year round, since the types of trails are not always interchangeable depending on how trails are being used).

For water and land trails our models project that the quickest rate of change will be for Type 2 users, averaged across Inuit Nunangat (see supplementary materials). We project that Type 2 users currently have access to water trails for 13 days/year and that the number of days will increase by 6 to 12 days/year (+46% and +92%) by 2100 under SSP245 and SSP585, respectively. On the other hand, we project that there are currently about 167 days/year that water trails are good for Type 3 users across Inuit Nunangat and that the number of good days will increase by 29% (49 days/year) and 63% (105 days/year) for low and high emissions scenarios, respectively.

Our models suggest that all user types will experience declining sea ice access, although the changes will affect Type 2 users the most (Fig. 5). We project that currently Type 2 users have 46 good days/year and that by the end of the century the number of good ice trail days will shrink to 27 days/year for Type 2 users under SSP245 (−41%) and 11 days/year (−76%) under SSP585. We project that Type 3 users currently have 172 good trail days/year, declining to 125 and 77 days/year by end of century under SSP245 and SSP585 (−27% and −55%), respectively.

## Discussion

**Translating model results into practice**. Overall, our models project relatively small declines in overall trail access across Inuit Nunangat, even at higher levels of global warming, although projected loss of access to sea ice trails is substantial. More important than these aggregate trends are how these changes will be experienced, perceived, and responded to in the context of how trails are used, for what purposes, and by whom. Thus, while we model an overall small decline in overall trail access, the impacts could be magnified if access is reduced to trails with critical importance to communities; such complexities of interaction require in-depth investigation in specific regions and communities. Several notable trends are evident along with opportunities for adaptation.

There is an unequal distribution of impacts projected by region and user type. Across trail types, we project that high-risk tolerance users (Type 3) on average presently have 324 days/year more access to at least one trail mode compared to a low-risk tolerance user (Type 2). We project that this gap will narrow to 308 days/year and 286 days/year days by 2100, varying by emission scenario. Our models suggest that Type 2 users will experience a slight increase in overall access.

Inequalities are particularly evident for sea ice trails, where the period at which such trails can be accessed becomes so short for Type 2 users that it could severely challenge their ability to develop the knowledge, skills, and confidence for using such trails, which is rooted in experiential learning[19,23]. In turn, this would make it more difficult to build the skills and experience

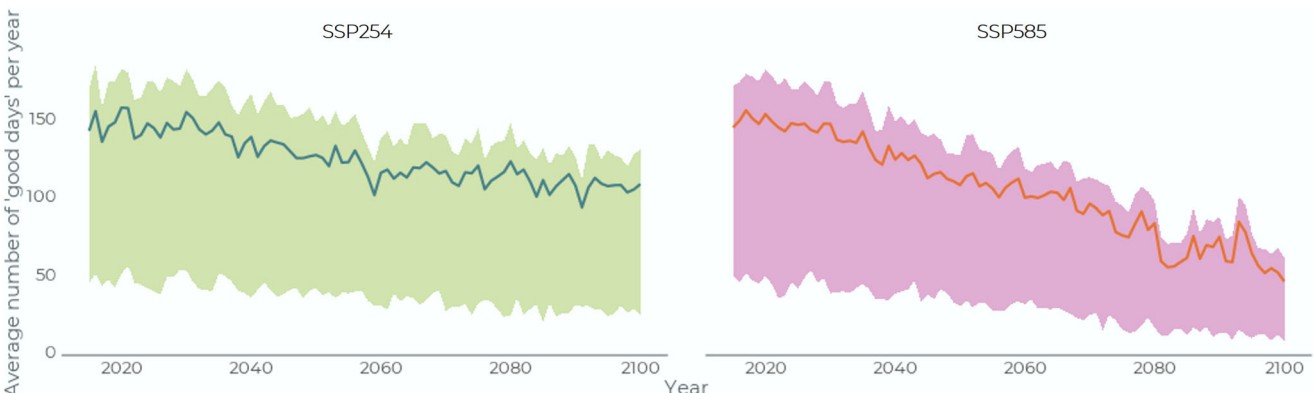

**Fig. 4 Regional trends in projected trail access.** Calculated for SSP245 and SSP585 emissions scenarios for type 1 trail user by region, 2015–2100 (circles around communities capture area covered by projections).

**Fig. 5 Annual number of good sea ice trail days.** Modeled for Type 1, 2, and 3 users, averaged across all communities for 2015–2100 for SSP245 and 585 (Type 1 users is the central line, type 2 is the lower line, type 3 the higher line).

reflected by the Type 3 user archetype, and difficult for expert knowledge to be retained and passed on between generations. Already with current climatic conditions in southern regions, Type 2 users have minimal access to ice trails and concerns have been expressed about the loss of associated skills for trail access[11,24]. In other regions, inter-annual variability is projected to overshadow longer-term trends in sea ice tail access initially, especially for SSP585 for Type 1 and 3 users, where projected changes largely remain within the bounds of access thresholds, with some years potentially seeing little change or an increase in access compared to previous years. However, by mid-century sea ice trail access is projected to change in all regions and for all categories of user. This has implications for adaptation: in previous work, for example, significant year-to-year variability has been identified to have made community members less inclined to invest in expensive equipment needed to take advantage of more open water conditions or manage more dangerous ice (e.g. boats, satellite phones)[25,26].

The seasonal profile of changing trail access has implications for the capacity to adapt. The literature documents how communities have altered trail modes in response to changing conditions, using more land and water-based trails as the ice has become more dangerous[21,27], with such responses having a long history of underpinning Inuit adaptability to environmental change[28,29].

Trail substitutability depends on a number of factors, and our projections raise concern over feasibility. Firstly, shifting trail modes requires an accessible alternative, with our models indicating that this is an unrealistic assumption with future climate change: for example, our models show that the periods when access to sea ice trails is projected to decline, land trails are challenging to use due to muddy conditions, with snow on the ground not deep enough for snowmobiles. Secondly, users need to have the competence and knowledge to use diverse trail modes. This is reasonable for Type 3 users, but for Type 1 and 2 users with lower skill sets, poorer equipment, and being more risk averse, our models demonstrate that improved water trail access will be unable to offset lost days of sea ice trail access (supplemental materials). For these reasons, trail substitutability may become more challenging in the future.

There are opportunities for adaptation through efforts to develop knowledge, skillsets, and improve access to equipment, although such opportunities decline at higher levels of warming. For example, if Type 2 users (low-risk tolerance) had the competence and confidence in travelling under the set of conditions of Type 1 users (average)—a not unreasonable goal —then trail access could increase by between 175 access days/year for SSP245 (307% increase) to 130 access days/year for SSP585 (164% increase) at end of century. Put another way, the number of days that someone had no trail access would be cut in half even under a high emissions scenario. This would extend the seasonality of access considerably for Type 2 users, who predominantly have modelled trail access in July and August compared to Type 1 users who maintain access to trails year-round across scenarios (Fig. 6). It would be particularly important for water trails, where Type 2 users have very few access days modelled this century, increasing access by 76 days/year under present conditions and between 98-122/year days by 2100. Programs focused on supporting skills development and learning following traditional Inuit ways have been a focus of adaptation initiatives across the Nunangat[30–32], and for the first time our models quantitatively support their importance in-light of climate warming.

Despite adaptation opportunities, limits to adaptation are projected in southern regions within the next 40 years for all types of trail user. This reflects the loss of access to sea ice trails, the associated socio-cultural importance of which cannot be replaced. In Nunatsiavut the SSP245 scenario can delay the passing of adaptation limits by approximately two decades, but even here ice thickness and concentration are projected to become too low for ice trails to be used (Fig. 7). In Nunavik models under the SSP245 scenario ice trail access is projected to stay relatively stable at the end of century compared to the 2060 s for Type 1 and 3 users, whereas the SSP585 scenario results in a loss of ice access for all users by the 2080 s. Limits to adaptation are not reached in other regions, although trail access for Type 2 users is projected to be significantly constrained at higher levels of warming. With a high level of warming, Type 1 and 3 users maintain access to all trail modes in five of seven regions modelled (access is not maintained in Nunavik and Nunatsiavut in 2090s), which is important given the role of such users in traditional food systems[29].

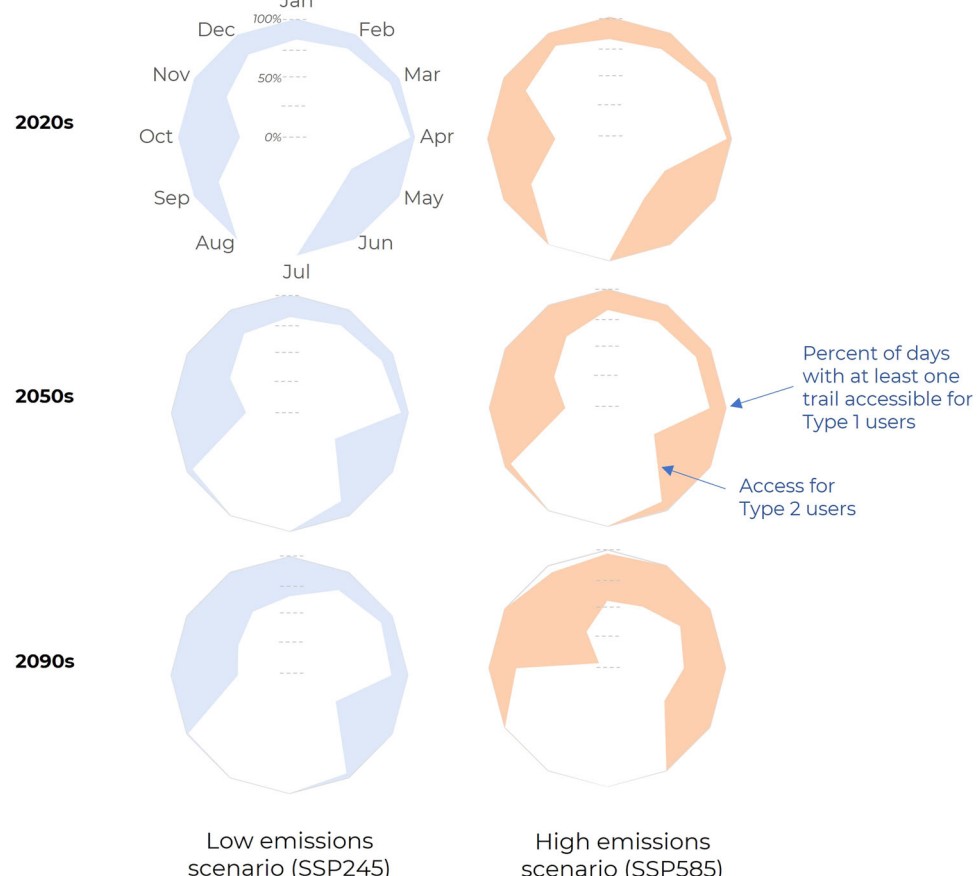

**Fig. 6 Monthly percentage of good days for Type 1 and Type 2 users for select decades under a high and a low emission scenario.** There is potential for adaptation to improve trail access under high and low emissions scenarios by giving Type 2 users (low-risk tolerance) the competence and confidence of Type 1 users (average).

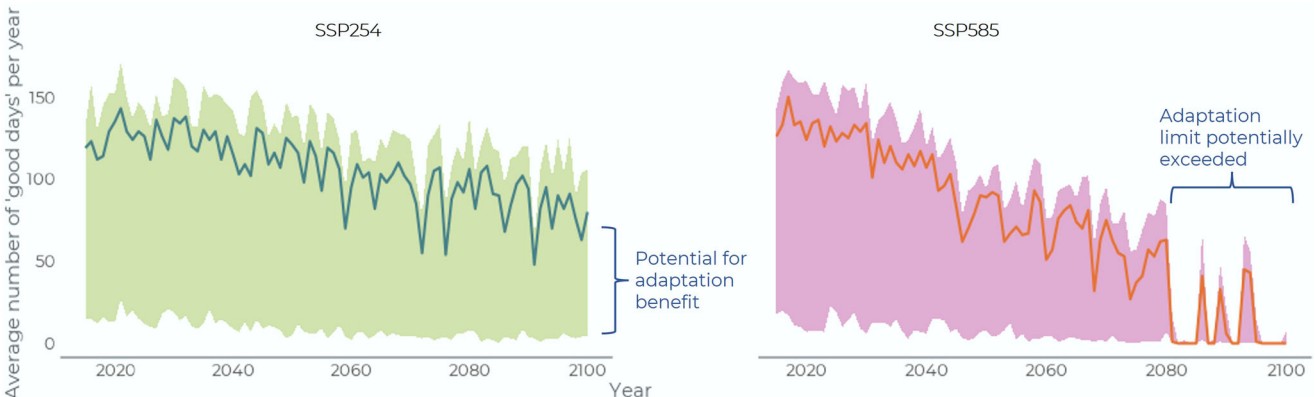

**Fig. 7 Annual number of good sea ice trail days.** Modeled for Type 1, 2, and 3 users for SSP245 and SSP585, averaged across Nunavik communities for 2015–2100, showing adaptation limits for SSP585 (Type 1 users is the central line, type 2 is the lower line, type 3 the higher line).

In teasing out these insights, we note that there are <u>large uncertainties in model output</u>, particularly for projected sea ice and wind conditions, but also temperature. GCMs produce wide-ranging estimates of future conditions with implications for modeled trail access (see supplementary materials). We manage these uncertainties by averaging across GCMs and using a high-emissions (SSP585) and low-emissions (SSP245) scenario, but model output nevertheless needs to be interpreted cautiously—capturing *potential* future trends and drivers of future risk that needs to be combined with alternative non-modeling-based approaches to envision the future. While some uncertainties are reducible through improved modeling, others are aleatoric and will always be present in seeking to project the future[3]. We also note the focus on modeling general associations across communities, which will mask some of the important complexities of how Inuit interact and respond to changing trail conditions. For example, the thresholds set for different types of user are designed to capture at a very general level how socio-economic characteristics might differentially affect trail access in-light of climate change; in real life these characteristics will interact and intersect in more complex ways, varying by trail type and location.

**Modelling climate futures**. By projecting how future warming might affect the ability to access trails, this paper compliments work on marine shipping, winter roads, and aviation[33–36] to develop a more comprehensive picture of how future climate change impacts might affect Arctic transport networks. The novel ethnoclimatology approach differs from standard climate impacts assessments, which typically focus on projecting changes in generalized climatic conditions rather than the specific conditions that matter to human livelihoods, health, and well-being. Yet, by accounting for differences in trail user characteristics due to skillsets, knowledge, and equipment—which can be thought of as proxies of sensitivity and adaptive capacity—we illustrate how changes in general climatic conditions by themselves only partially explain potential future impacts. Except for Nunatsiavut under both high and low emissions scenarios, and Nunavik and Nunatsiavut under the high emissions scenario where limits to adaptation are projected due to a loss of usable sea ice, it is how changing conditions interact with trail user characteristics that determines the magnitude, spatial distribution, and social impacts of projected changes, even at levels of warming up to +8.6 °C above present.

The regional focus of this paper responds to calls from decision makers to provide input on how climate change might affect trail use and opportunities for adaptation across Inuit Nunangat[37–40].

Our projections build upon the knowledge of trail users themselves but nevertheless we note that 'scaling-up' regionally involves generalisations of trail access trends where local context is lost. Moreover, the risk tolerance thresholds in our models remain constant over time, overlooking the potential for adaptive learning or the accumulation of vulnerability as climate change impacts manifest[41]. For these reasons, it is important that the results are interpreted as providing broad-level insights of potential future trends. In future research, we will address some of these challenges by developing trail access models specific to selected communities and as part of participatory scenario planning processes that explore how projected trends may be experienced and responded to locally.

## Methods

**Modelling framework**. Human dimensions of climate change research across the Arctic is dominated by two distinct and often separate approaches. In the first, scientists use climate projections to examine how infrastructure or communities may be affected by shifting hazards over time, with the goal of identifying and quantifying risks directly linked to climate change. In the second, the focus is on cataloguing local knowledge and Indigenous knowledge, observing people's routines and participating in elements of daily life, seeking to characterise the relationship people have with the environment and understand how social processes interact with environmental risks. The strengths and weaknesses of these approaches are reviewed elsewhere[42–44], with Western systems of government that dominate Arctic states and regional decision-making frequently relying on quantitative evidence provided by the first approach, overlooking the importance of socio-cultural factors outlined in more qualitative work[45].

Our ethnoclimatology modelling framework bridges this gap by weaving the knowledge of trail users and climate change modelling together, focusing on climatic conditions identified to be important to trail users themselves and capturing social factors through a focus on different categories of trail user. This process allows us to 'translate' physical data into a socially meaningful measure[46]. The framework has seven steps (Fig. 8), with steps one to four reported in detail in Ford et al.[22]. Additional details on the methodology can be found in the supplemental materials.

- **Step 1: semi-structured interviews** ($n = 273$) were conducted with regular trail users, with the aim of developing a generalizable understanding of climate-relevant conditions affecting trail access across Inuit Nunangat. Interview questions focused on documenting knowledge about past and current use of trails, the nature of climate-related conditions posing risks, and how risks are perceived and managed. These interviews and the consent process are documented in Ford et al.[22], with questions also allowing us to assess how trail access differs by individual depending on environmental knowledge and skill sets, access to resources, and risk tolerance. To contextualize our qualitative findings, we employed methods of triangulation, member-checking, ground truthing, and spending considerable time traveling with trail users across seasons from 2015–2017, asking questions while using trails. The communities were selected to capture a sample reflective of diverse settlements and the varied geographies in which trails are used, with the aim of developing a generalizable understanding of climate-relevant conditions affecting trail access across Inuit Nunangat. Team members had well-established working relationships with the selected communities prior to this project

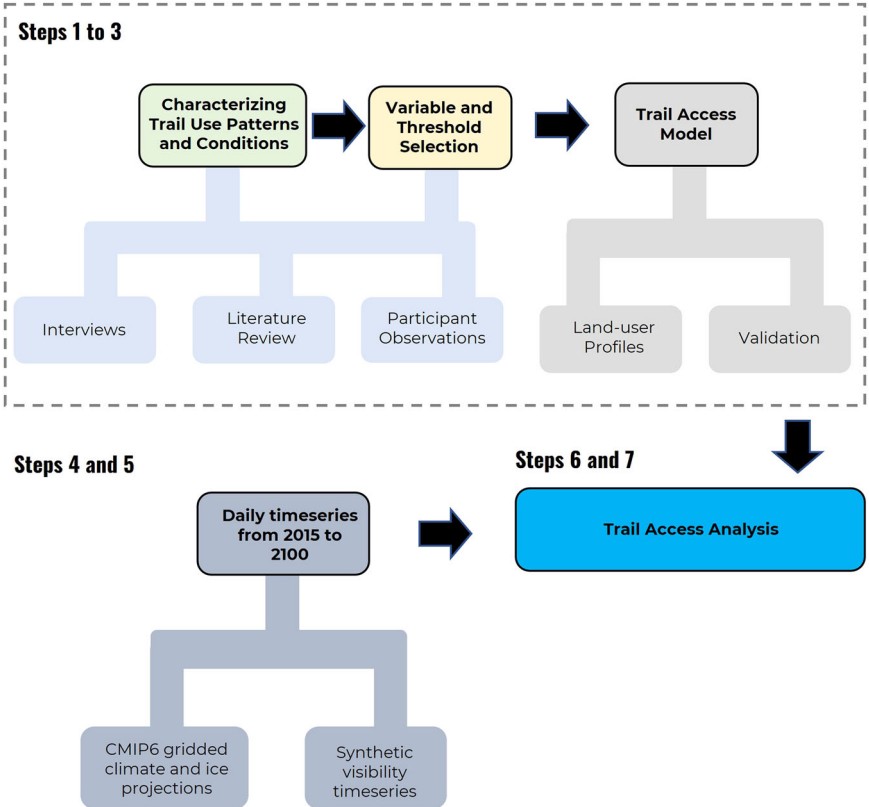

**Fig. 8 Steps used in the modelling framework.** The framework is composed of 7 steps, beginning by identifying and characterising climatic conditions that matter for trail access.

commencing, with the work conducted under a Nunavut Research Institute License, Aurora Research Institute Scientific Research License, with Human Research Ethics Approval from McGill University and the University of Guelph.

- **Step 2: trail use thresholds** were created by translating qualitative data (Step 1) into thresholds for climate and ice conditions. This involved developing a list of variables specific to each climate-related condition that could be measured, which define whether a trail can be used or not, focusing on three trail modes. To account for variation in knowledge, skills sets, equipment, and risk tolerance of trail users, thresholds were set differently for different categories of trail user: Type 1 (average risk tolerance); Type 2 (low-risk tolerance); and Type 3 (high-risk tolerance)[22]. Thresholds were identified by analysing interview transcripts, disaggregated by trail type and user category, with interviewees explicitly asked about specific thresholds that limit trail access; thresholds were also imputed from interviewee descriptions of 'good' and 'bad' conditions[22]. Thresholds are documented in Table S2.
- **Step 3: trail access models were** developed that could characterize trail access on a particular day, whereby each variable was classed as a "pass" or a "fail" on a specific day using the thresholds for each trail mode and user type. Passes and fails were then aggregated and, if >15% of variables were classed as a fail, the trail was defined as not accessible on the particular day in question[22].
- **Step 4: a visibility time series was derived** from historic climate data from each region. Because visibility is not projected by climate models we created a synthetic daily time series by randomly sampling 20 years of historic visibility data for each region. We compared distributions of the historic data with the synthetic time series in order to ensure they were not significantly different.
- **Step 5: daily time series of climate and sea ice projections** from 2015 to 2100 were developed using CMIP6 gridded climate data. A daily timeseries was developed for five GCMs (BCC-CSM2-MR; CMCC-ESM2; CMCC-CM2-SR5; MRI-ESM2-0; and NorESM2-MM) and two emissions scenarios (SSP245 and SSP585). All GCMs selected offered gridded data at a 100 km resolution for all variables. Daily temperature (tas), precipitation (pr), and surface wind (sfcWind) were calculated based on the average of all GCM grid cells within 100 km radius of each of the Inuit Nunangat's 53 communities. Sea ice concentration (siconc) and sea ice thickness (sithick) were calculated using average grid values for a 150 km radius around each

community (supplemental materials). The daily climate and ice projections were then linked up with the daily visibility time series.

- **Step 6: the number of 'pass' and 'fail' days** for each GCM and emissions scenario was calculated by applying the models from Step 3 to the time series from Step 5. Because of the shift from point-based data described in the historical analysis[22] to the gridded climate and ice data, we iteratively calibrated our models' thresholds (Step 3) to achieve a difference between the historic time series (2010 to 2016) and projections (2015 to 2020 average) of $+-30\%$ (supplemental materials).
- **Step 7: Analysis of temporal and spatial patterns** was done through descriptive statistics. Generally, we assessed the difference between average values for 2015 to 2030 compared to the average values for 2090 to 2100.

**Reporting summary.** Further information on research design is available in the Nature Portfolio Reporting Summary linked to this article.

## Data availability
The datasets generated during and/or analysed during the current study are available online at a publicly accessible website: https://climatechoices.shinyapps.io/ArcticTrack/ The thresholds used in the trail access models are available in Supplementary Table S2.

## Code availability
The R scripts developed to synthesize and analyze climate data for this study are available from GitHub, "https://github.com/dylangclark/ArcticTrack".

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

## Acknowledgements

We would like to thank Jackie Dawson at the University of Ottawa for sea ice chart conversion. We received funding from the Canadian Institutes of Health Research, the Natural Sciences and Engineering Research Council of Canada, ArcticNet Network of Centres of Excellence Canada, University of Leeds, and the University of Ottawa. We thank three anonymous reviewers who provided constructive feedback. We thank all the community members who were involved in the original research from which this paper builds.

## Competing interests

The authors declare no competing interests.

## Additional information

## IHACC Research Team

James D. Ford[1], Sherilee Harper[5], Lea Berrang Ford[1], Cesar Carcamo[6], Patricia Garcia[6], Shuaib Lwasa[7], Didacus Namanya[8], Mark New[9] & Carol Zavaleta-Cortijo[6]

[6]Universidad Peruana Cayetano Heredia, Lima, Peru. [7]Erasmus University, Rotterdam, Netherlands. [8]Ministry of Health, Kampala, Uganda. [9]University of Cape Town, Cape Town, South Africa.

