## [Peer Review File · Communications Earth & Environment]

9th Aug 22

Dear Professor Ford,

Your manuscript titled "Projected changes to trail access in the Arctic" has now been seen by 3 reviewers, and I include their comments at the end of this message. They find your work of interest, but some important issues are raised. We are interested in the possibility of publishing your study in *Communications Earth & Environment*, but would like to consider your responses to these concerns and assess a revised manuscript before we make a final decision on publication.

We therefore invite you to revise and resubmit your manuscript, along with a point-by-point response that takes into account the points raised. Please highlight all changes in the manuscript text file.

Your study covers an important area of our journal. However, readers in the field of natural science may have difficulty with your text because the methodology is based on the previous publication as well as the social aspect of the study. Therefore, please revise the manuscript in consideration of a broad range of journal readers. Let me remind you that our format allows up to 5000 words for main text and unlimited space for methods.

Please use the following link to submit your revised manuscript, point-by-point response to the referees' comments (which should be in a separate document to any cover letter) and the completed checklist:

[link redacted]

We hope to receive your revised paper within six weeks; please let us know if you aren't able to submit it within this time so that we can discuss how best to proceed. If we don't hear from you, and the revision process takes significantly longer, we may close your file. In this event, we will still be happy to reconsider your paper at a later date, as long as nothing similar has been accepted for publication at *Communications Earth & Environment* or published elsewhere in the meantime.

We understand that due to the current global situation, the time required for revision may be longer than usual. We would appreciate it if you could keep us informed about an estimated timescale for resubmission, to facilitate our planning. Of course, if you are unable to estimate, we are happy to accommodate necessary extensions nevertheless.

Please do not hesitate to contact me if you have any questions or would like to discuss these revisions further. We look forward to seeing the revised manuscript and thank you for the opportunity to review your work.

Best regards,

Shin Sugiyama, PhD
Editorial Board Member
Communications Earth & Environment
orcid.org/0000-0001-5323-9558

Clare Davis, PhD
Senior Editor
Communications Earth & Environment

EDITORIAL POLICIES AND FORMATTING

Editorial Policy: [Policy requirements](https://www.nature.com/documents/nr-editorial-policy-checklist.zip)

Furthermore, please align your manuscript with our format requirements, which are summarized on the following checklist:

[Communications Earth & Environment formatting checklist](https://www.nature.com/documents/commsj-phys-style-formatting-checklist-article.pdf)

and also in our style and formatting guide [Communications Earth & Environment formatting guide](https://www.nature.com/documents/commsj-phys-style-formatting-guide-accept.pdf) .

***** DATA:** Communications Earth & Environment endorses the principles of the Enabling FAIR data project (<http://www.copdess.org/enabling-fair-data-project/>). We ask authors to make the data that support their conclusions available in permanent, publically accessible data repositories. (Please contact the editor if you are unable to make your data available).

All Communications Earth & Environment manuscripts must include a section titled "Data Availability" at the end of the Methods section or main text (if no Methods). More information on this policy, is available at <http://www.nature.com/authors/policies/data/data-availability-statements-data-citations.pdf>.

If a community resource is unavailable, data can be submitted to generalist repositories such as [figshare](https://figshare.com/) or [Dryad Digital Repository](http://datadryad.org/). Please provide a unique identifier for the data (for example a DOI or a permanent URL) in the data availability statement, if possible. If the repository does not provide identifiers, we encourage authors to supply the search terms that will return the data. For data that have been obtained from publically available sources, please provide a URL and the specific data product name in the data availability statement. Data with a DOI should be further cited in the methods reference section.

REVIEWER COMMENTS:

Reviewer #1 (Remarks to the Author):

This manuscript presents the results of an analysis of trail access conditions in Inuit Nunangat based on a comparison of climate models and characteristics of trail use and trail users in the region. This study extends to the future work that Ford et al. have done looking at past changes (reference 19 in this manuscript). The work is well done, insightful, and an important contribution to understanding of the implications of climate change in the Arctic. The assessment of trail user characteristics is especially important in identifying divergent effects as well as the potential for adaptation through improved skill and knowledge.

I see some areas for improvement in the presentation of the study and its findings. I realize that word limits may restrict how much detail the authors can add, but readers unfamiliar with the Arctic and with the authors' previous work may have a harder time following the presentation. Some details can be added to the Supplemental Materials if needed, but the more that can be put in the main text, the better.

Lines 69-71: Types 1-3 are described here in terms of risk tolerance, but risk tolerance is listed as just one of several characteristics that are combined into the Types. For example, someone skilled and knowledgeable in boat travel may have far less skill and knowledge in snowmobile travel, but the same level of risk tolerance in both. The outcomes will be different, since in the former case the skill and knowledge will allow much greater access even though the risk tolerance by itself has not changed. Perhaps choose a different term for the aggregate measure. Note too that Table S1 refers to “skill” rather than risk tolerance with regard to the three Types. Skill, too, is only one of the components of travel competence.

70-71: Type 1, Type 2, and Type 3 do not proceed sequentially, but move from Average to Low to High. This becomes confusing later on when the terms are used again. Perhaps avoid using numbers, and just refer to Low, Average, and High?

88-95: The number of days below -30C is interesting, but wouldn't a threshold of 0C or a measure such as freezing degree days be more relevant? I'm not sure of the significance of cold days here, other than as one way of indicating temperature change over time.

Figure 2, panels c and d: how meaningful is the annual average of ice conditions? I realize again that this is an indicator, but wouldn't winter conditions be more relevant? At least some explanation of how annual averages compare to conditions during sea ice travel season would be welcome.

132ff: I could not find in the main text or the supplementary materials any details about what constitute good travel conditions for water, land, or ice. This could be provided in a table or in an additional paragraph. Without some description, much of the analysis is hard to follow. It is also unclear, for example looking at Figure 3, why land travel is not high in winter. Does land travel exclude snowmobiles? Is “ice” referring mainly to sea, river, and lake ice, or does it mean snow on land as well? Other details are also not clear, for example why wind affects land travel (lines 166-7). Does this refer to blowing snow, or something else? This is perhaps the biggest weakness of the current presentation, that there is no clear description of what constitutes good trail conditions and why. Without that information, readers are left taking much of the analysis on faith.

213-4: The idea that high-competence users have year-round access to at least one type of trail is an interesting one. That doesn't mean, though, that they can get everywhere all year round, since the types of trails are not interchangeable. It might be worth adding a brief commentary on this point. See also lines 239-40, which imply that trail access won't change much. True overall, but it may mean that some areas remain accessible whereas others become harder or even impossible to reach. I know this is beyond the scope of the study, but I suggest at least mentioning it along with the other limitations and implications of the analysis presented here.

Supplementary Materials

“Difference by climate models (GCMs)” uses Celsius, whereas Table S1 used Kelvin. This makes the two hard to compare directly. I realize the models may use K, but is that necessary here? Especially in Table S1, using K means most readers will have to do a mental conversion to see what the temperatures mean.

Table S1: I may not be reading this correctly, but some of the entries don't make sense to me. Some of my confusion may come from the headings "Fail>X" and "Fail<X". I read those to mean, respectively, that if the parameter is greater than X it's a fail and if it's lower than X it's a fail. If that's so, though, then it would appear that the High Skill user of Land trails has a narrower temperature window than the other users (first three rows of the table). Is that true? If so, some explanation would be nice. Also, the High Skill user appears to need more snow for travel. Similarly, the entries for wind appear to say that winds below (rather than above) a certain threshold are a "fail" which does not make sense to me. If I am misreading the table my apologies, but clearer labeling would be helpful. Here, again, it would be helpful to have a description of how the different parameters affect trail use, so that readers can make sense of the chart. For example, the mention of muddy trails suggests that precipitation is a factor at least for summer land trail use, but that is only mentioned in passing rather than as an important parameter of summer trails on land.

Reviewer #2 (Remarks to the Author):

A very interesting and thorough treatment of an important question on the relationship between climate change and movement in the arctic. Rapid changes to sea ice conditions, animal movements, and acceleration of geomorphological processes all impact the character of choices and timing through which trips are made. As the authors note, movement is a critical component of connection between communities.

I think the interview process, application CMIP6 projections, and nuanced consideration of implications for trails is appropriate and well discussed.

My only suggestion is in framing the importance of routes of movement from a cultural perspective. The authors cite Aporta, Simonee and Gearheard's works, for example, but still leave the reader with a sense that it is still just a matter of "transit" under potential threat. Movement is also a critical component of perception, and a fundamental part of how Inuit communities apprehend change. Routes define the circumstances through which animals are encountered and the scenarios through which humans and animals behave in relation to each other. Routes are also a connection to the past, and often follow archaeological traces, which are themselves being impacted by climate change related processes - which in some cases are quite extreme (e.g. Walls et al 2020). If routes of movement part of the direct perceptual experience of change, they are also part of how communities respond creatively with environmental change.

Walls, M., Hvidberg, M., Kleist, M., Knudsen, P., Mørch, P., Egede, P., ... & Watanabe, T. (2020). Hydrological instability and archaeological impact in Northwest Greenland: Sudden mass movement events signal new concerns for circumpolar archaeology. *Quaternary Science Reviews*, 248, 106600.

Reviewer #3 (Remarks to the Author):

This article offers an innovative approach to studying how climate change will impact over time the capacity of Inuit to travel over three kinds of traditional routes: sea ice trails, open water routes, and inland trails. Each of these represents a different type of landscape corresponding to traditional seasonal patterns of mobility. The authors argue very reasonably that access to these routes will be differential. Some people will be impacted more than others; some routes will be materially impacted more than others; and so, understanding and anticipating these differential impacts could help different stakeholders, particularly the Inuit people who travel these trails, to prepare and adapt to these changes. Within the general rubric of human ecology and climate change impacts, the study is logical, coherent, and meets a need to develop tools for adaptation planning. Moreover, the attention to differential impacts recognises the complexity of a research project of this kind and the challenge of producing robust and reliable conclusions. Overall, then, here is a project that is innovative, sitting at the intersection of social and natural sciences, challenging by virtue of the complexity of Inuit seasonality and mobility, as well as climate models.

One of the clear virtues of this manuscript is its collaborative, interdisciplinary framework (ethnoclimatology) and methodology, reflected in its multiple authorships, institutions, and disciplinary formations. This is very much welcomed.

The authors pay close attention to changing patterns of precipitation and wind, as well as temperature, all of which are very important for mobility and travel. The use of five Global Circulation Models, together with three categories of risk tolerance, and land, water and sea ice “trail types” gives the authors plenty of scope for exploring impacts. This matrix is at the heart of the paper and is in effect the engine producing the different sets of projected access times.

One quandary to highlight as it is as important for the journal editors as for the authors is that the strength of the article rests on the use of thresholds to define user types. The authors tend to assume that readers of this manuscript are familiar with the explanations and justification of the user types developed in an article by Ford, Clark, Pearce et al (2019) in *Nature Climate Change*. In that context this article can be seen as a further development. That is understandable, and even a very good idea, but it does mean that important parts of the 2019 article are so condensed here that to a certain extent one has to take them on faith (difficult to do as a peer reviewer). Specifically the definitions (if I understand correctly) are mainly set out in Table 1 of the 2019 article, but not reproduced or summarised here, so this reader was left guessing. Very importantly for this purpose, the thresholds are based on users’ judgements about their competence or risk aversion in relation to temperature, precipitation, wind and visibility. That is crucial but it is not clear enough in this article. One reason to be more explicit is that there are a sizeable number of competing theories of risk, including anthropological studies of risk in Inuit culture, where other logics, rationalities, and factors play important roles.

In that vein, factors listed such as “knowledge and skillsets..., financial resources, and access to technology” are not necessarily correlated or aligned in Inuit communities. Some people have great knowledge but limited financial resources for example. That being the case, it is hard to know what determines whether someone is a low risk (2) or high risk (3) user – or whether the category of user is really just a threshold where it is understood that the range

of socio-cultural factors is varied and complex sitting in the background. That vagueness can be seen as masking the intersectionality of cultural factors at play (though that is presumably not the authors' intention as they acknowledge and even emphasise the complexity of the study).

The research findings are interesting, particularly where they might be seen to be surprising and not self-evident. Amongst these are the projects of "relatively small declines in overall trail access", pointing to the importance of the more granular conclusions.

Interannual variability is recognised as it "may mask some of these trends in the short term, but by mid-century trail access is projected to change in all regions and for all categories of users". This would benefit from some clarifications for three reasons. One is that interannual variability is the subject of ongoing studies and not easy to specify with confidence. Second, the sentence quoted above is confusing as it seems to be trying to say two or three things at once. Are the authors saying that interannual variability is largely independent of trail access come mid-century, one signal at times masking the other, but that this will stop mid-century? Thirdly recent and current research suggest that the kind of variability being widely described today as "extreme weather events" could become a dominant driver impacting many peoples and communities in the northern hemisphere, including northern indigenous peoples. Clarifying this paragraph about impacts and different kinds of variability would strengthen the article.

The study as well as projecting trail access declining, also highlights that the gap between different kinds of users (thresholds) is likely to decline. That is very interesting, but the findings need presenting more systematically. One is not surprised to learn that sea ice access and skills in Nunatsiavut are most under threat given its southerly location and current climatic and oceanographic conditions. However the reader is given a sufficiently full picture across the other regions or sub-regions. Compressing this information into a short paragraph fails to convey the bigger picture across Nunangat. That said, there are good insights such as reinforcing the fact that trail types are not easily substituted for one another. That has pretty much always been the case with sea ice, open water, and land for many reasons, though perhaps the argument here is that this will be even more true in the future?

In summary, the project is a good one in principle, as is the collaborative approach. I would like to see the authors clarify how this article connects to, and builds on, previous work where key principles around thresholds and trail types were developed. Having now read that article, this submitted article becomes much more clear. That being the case, I can see how certain findings are highlighted, but I would like to see a more systematic presentation of results, a fuller discussion of climate variability, some discussion of extreme weather events, and some further probing of the intersectional issues in the discussion of adaptation strategies. Finally some analysis of the reliability of the projections would strengthen the article to justify rounding percentage change in trail access to the nearest one per cent, which looks over-optimistic.

Dear editor,

Thank you for the constructive feedback on our paper "Projected changes to trail access in the Arctic." We respond to each comment in the table below and believe that the paper has been strengthened with these changes.

Reviewer 1	
This manuscript presents the results of an analysis of trail access conditions in Inuit Nunangat based on a comparison of climate models and characteristics of trail use and trail users in the region. This study extends to the future work that Ford et al. have done looking at past changes (reference 19 in this manuscript). The work is well done, insightful, and an important contribution to understanding of the implications of climate change in the Arctic. The assessment of trail user characteristics is especially important in identifying divergent effects as well as the potential for adaptation through improved skill and knowledge. I see some areas for improvement in the presentation of the study and its findings. I realize that word limits may restrict how much detail the authors can add, but readers unfamiliar with the Arctic and with the authors' previous work may have a harder time following the presentation. Some details can be added to the Supplemental Materials if needed, but the more that can be put in the main text, the better.	Thank you. We are pleased you like the paper
Lines 69-71: Types 1-3 are described here in terms of risk tolerance, but risk tolerance is listed as just one of several characteristics that are combined into the Types. For example, someone skilled and knowledgeable in boat travel may have far less skill and knowledge in snowmobile travel, but the same level of risk tolerance in both. The outcomes will be different, since in the former case the skill and knowledge will allow much greater access even though the risk tolerance by itself has not changed. Perhaps choose a different term for the aggregate measure. Note too that Table S1 refers to "skill" rather than risk tolerance with regard to the three Types. Skill, too, is	We agree that the user archetypes we create for the model do not capture all possible combinations of skills, risk aversion, equipment, and knowledge. Through the three archetypes we simply aim to articulate a continuum of possible impacts for each trail type. We have worked to describe this more precisely in the manuscript and we have also added content to the supplemental materials which emphasizes the various components, or factors, which may place someone in the Type 1, 2, or 3 group. We have, however, elected to keep the mention or risk aversion in line 76, simply as a means of introducing the three archetypes. We have worked to use other terms where possible throughout other section of the manuscript.

only one of the components of travel competence.	
70-71: Type 1, Type 2, and Type 3 do not proceed sequentially, but move from Average to Low to High. This becomes confusing later on when the terms are used again. Perhaps avoid using numbers, and just refer to Low, Average, and High?	We appreciate the reviewer's comments and have made a few adjustments to improve readability and clarity. In select places throughout the document, we have made sure to note that Type 1 users are the "average" archetype. We have also added a table to the supplemental materials (S1) as an easy reference for readers about what each archetype represents.
88-95: The number of days below -30C is interesting, but wouldn't a threshold of 0C or a measure such as freezing degree days be more relevant? I'm not sure of the significance of cold days here, other than as one way of indicating temperature change over time.	The reviewer makes a good observation about the selection of -30C as a threshold in line 98. We chose -30 as it is more reflective of a threshold for rapid sea ice development. Further, we observe that temperatures are shifting most rapidly at these important 'tails'. In response to this comment and other reviewer comments, we have created an interactive online portal (https://climatechoices.shinyapps.io/ArcticTrack/) that will allow readers to further explore the data. This includes examining temperature thresholds as well as distributions of all the climate variables we use in this study.
Figure 2, panels c and d: how meaningful is the annual average of ice conditions? I realize again that this is an indicator, but wouldn't winter conditions be more relevant? At least some explanation of how annual averages compare to conditions during sea ice travel season would be welcome	The reviewer is correct that winter ice conditions are most relevant to trail use; however, since this is the first paper to our knowledge that examines CMIP6 projections of coastal sea ice across Northern Canada, we believe that it is important to start with a high-level picture of the broad shifts projected. Average ice conditions are an illustrative indicator of changes to ice in coastal waters across Inuit Nunangat, and it is in the next section where we develop more specific projections of how climatic conditions that matter will change.
132ff: I could not find in the main text or the supplementary materials any details about what constitute good travel conditions for water, land, or ice. This could be provided in a table or in an additional paragraph. Without some description, much of the analysis is hard to follow. It is also unclear, for example looking at Figure 3, why land travel is not high in winter. Does land travel exclude snowmobiles? Is "ice" referring mainly to sea, river, and lake ice, or does it mean snow on land as well? Other details are also not clear, for example why wind affects	Thank you for noting this comment. Based on reviewers' feedback we have added additional information to both the supplemental materials as well as the main manuscript. We have added additional details about why the thresholds were selected, what their significance is to Inuit trail users, and additional literature that supports the selection of the variable or criteria. The reviewer also makes an important observation about Figure 3, which was not clearly conveying its intended point. We have replaced

land travel (lines 166-7). Does this refer to blowing snow, or something else? This is perhaps the biggest weakness of the current presentation, that there is no clear description of what constitutes good trail conditions and why. Without that information, readers are left taking much of the analysis on faith.	the figure with a graphic that more clearly communicates the results.
213-4: The idea that high-competence users have year-round access to at least one type of trail is an interesting one. That doesn't mean, though, that they can get everywhere all year round, since the types of trails are not interchangeable. It might be worth adding a brief commentary on this point. See also lines 239-40, which imply that trail access won't change much. True overall, but it may mean that some areas remain accessible whereas others become harder or even impossible to reach. I know this is beyond the scope of the study, but I suggest at least mentioning it along with the other limitations and implications of the analysis presented here.	This is a good point and we have added a note to this effect in the paragraph in question and the discussion.
Supplementary Materials "Difference by climate models (GCMs)" uses Celsius, whereas Table S1 used Kelvin. This makes the two hard to compare directly. I realize the models may use K, but is that necessary here? Especially in Table S1, using K means most readers will have to do a mental conversion to see what the temperatures mean.	Thank you. This is a good observation of a way to improve clarity. We have made the suggested changes to the table.
Table S1: I may not be reading this correctly, but some of the entries don't make sense to me. Some of my confusion may come from the headings "Fail>X" and "Fail<X". i read those to mean, respectively, that if the parameter is greater than X it's a fail and if it's lower than X it's a fail. If that's so, though, then it would appear that the High Skill user of Land trails has a narrower temperature window than the other users (first three rows of the table). Is that true? If so, some explanation would be nice. Also, the High Skill user appears to need more snow for travel. Similarly, the entries for wind appear to say that winds below (rather than above) a certain threshold are a "fail" which does not make sense to me. If i am misreading the table my apologies, but clearer labeling would be helpful. Here, again, it would be	Thank you. This is a good observation of a way to improve clarity. We have made changes to the table (S2) to improve clarity and ease interpretation. We believe the changes will clear up confusion about what the metrics mean and how they translate to trail use conditions.

helpful to have a description of how the different parameters affect trail use, so that readers can make sense of the chart. For example, the mention of muddy trails suggests that precipitation is a factor at least for summer land trail use, but that is only mentioned in passing rather than as an important parameter of summer trails on land.	
Reviewer 2	
A very interesting and thorough treatment of an important question on the relationship between climate change and movement in the arctic. Rapid changes to sea ice conditions, animal movements, and acceleration of geomorphological processes all impact the character of choices and timing through which trips are made. As the authors note, movement is a critical component of connection between communities. I think the interview process, application CMIP6 projections, and nuanced consideration of implications for trails is appropriate and well discussed.	Thank you. We are pleased you like the paper
My only suggestion is in framing the importance of routes of movement from a cultural perspective. The authors cite Aporta, Simonee and Gearheard's works, for example, but still leave the reader with a sense that it is still just a matter of "transit" under potential threat. Movement is also a critical component of perception, and a fundamental part of how Inuit communities apprehend change. Routes define the circumstances through which animals are encountered and the scenarios through which humans and animals behave in relation to each other. Routes are also a connection to the past, and often follow archaeological traces, which are themselves being impacted by climate change related processes - which in some cases are quite extreme (e.g. Walls et al 2020). If routes of movement part of the direct perceptual experience of change, they are also part of how communities respond creatively with environmental change.	Thank you for this comment. We have expanded our description in the intro on the cultural importance of trail use to tease out the deep connections between trails and Inuit culture, history, and identity.
Reviewer 3	

This article offers an innovative approach to studying how climate change will impact over time the capacity of Inuit to travel over three kinds of traditional routes: sea ice trails, open water routes, and inland trails. Each of these represents a different type of landscape corresponding to traditional seasonal patterns of mobility. The authors argue very reasonably that access to these routes will be differential. Some people will be impacted more than others; some routes will be materially impacted more than others; and so, understanding and anticipating these differential impacts could help different stakeholders, particularly the Inuit people who travel these trails, to prepare and adapt to these changes. Within the general rubric of human ecology and climate change impacts, the study is logical, coherent, and meets a need to develop tools for adaptation planning. Moreover, the attention to differential impacts recognises the complexity of a research project of this kind and the challenge of producing robust and reliable conclusions. Overall, then, here is a project that is innovative, sitting at the intersection of social and natural sciences, challenging by virtue of the complexity of Inuit seasonality and mobility, as well as climate models.

One of the clear virtues of this manuscript is its collaborative, interdisciplinary framework (ethnclimatology) and methodology, reflected in its multiple authorships, institutions, and disciplinary formations. This is very much welcomed.

The authors pay close attention to changing patterns of precipitation and wind, as well as temperature, all of which are very important for mobility and travel. The use of five Global Circulation Models, together with three categories of risk tolerance, and land, water and sea ice “trail types” gives the authors plenty of scope for exploring impacts. This matrix is at the heart of the paper and is in effect the engine producing the different sets of projected access times.

Thank you. We are pleased you like the paper

One quandary to highlight as it is as important for the journal editors as for the authors is that the strength of the article rests on the use of thresholds to define user types. The authors tend to assume that readers of this manuscript are familiar with the explanations and justification of the user types developed an article by Ford, Clark, Pearce et al (2019) in Nature Climate Change. In that context this article can be seen as a further development. That is understandable, and even a very good idea, but it does mean that important parts of the 2019 article are so condensed here that to a certain extent one has to take them on faith (difficult to do as a peer reviewer). Specifically the definitions (if I understand correctly) are mainly set out in Table 1 of the 2019 article, but not reproduced or summarised here, so this reader was left guessing. Very importantly for this purpose, the thresholds are based on users' judgements about their competence or risk aversion in relation to temperature, precipitation, wind and visibility. That is crucial but it is not clear enough in this article. One reason to be more explicit is that there are a sizeable number of competing theories of risk, including anthropological studies of risk in Inuit culture, where other logics, rationalities, and factors play important roles.	We have added more information as requested in the introduction to the paper and in the methods section. We have also added a table with the thresholds captured (S2)
In that vein, factors listed such as “knowledge and skillsets..., financial resources, and access to technology” are not necessarily correlated or aligned in Inuit communities. Some people have great knowledge but limited financial resources for example. That being the case, it is hard to know what determines whether someone is a low risk (2) or high risk (3) user – or whether the category of user is really just a threshold where it is understood that the range of socio-cultural factors is varied and complex sitting in the background. That vagueness can be seen as masking the intersectionality of cultural factors at play (though that is presumably not the authors' intention as they acknowledge and even emphasise the complexity of the study).	These are good points. We note that, like all modeling our approaches, our work involves some degree of simplification and generalization. We have added extra detail in the discussion and methods sections on the approach used, and note limitations of such simplifications (including, as the reviewer notes here around the specification of thresholds by trail user).
The research findings are interesting, particularly where they might be seen to be surprising and not self-evident. Amongst	Thanks for the comment

these are the projects of “relatively small declines in overall trail access”, pointing to the importance of the more granular conclusions.	
Interannual variability is recognised as it “may mask some of these trends in the short term, but by mid-century trail access is projected to change in all regions and for all categories of users”. This would benefit from some clarifications for three reasons. One is that interannual variability is the subject of ongoing studies and not easy to specify with confidence. Second, the sentence quoted above is confusing as it seems to be trying to say two or three things at once. Are the authors saying that interannual variability is largely independent of trail access come mid-century, one signal at times masking the other, but that this will stop mid-century? Thirdly recent and current research suggest that the kind of variability being widely described today as “extreme weather events” could become a dominant driver impacting many peoples and communities in the northern hemisphere, including northern indigenous peoples. Clarifying this paragraph about impacts and different kinds of variability would strengthen the article.	We have removed this sentence, and instead focus in more depth on what interannual variability looks like in a sea ice context where its impacts are most noticeable, and add extra text to explain.
The study as well as projecting trail access declining, also highlights that the gap between different kinds of users (thresholds) is likely to decline. That is very interesting, but the findings need presenting more systematically. One is not surprised to learn that sea ice access and skills in Nunatsiavut are most under threat given its southerly location and current climatic and oceanographic conditions. However the reader is given a sufficiently full picture across the other regions or sub-regions. Compressing this information into a short paragraph fails to convey the bigger picture across Nunangat.	We note here that our focus is on Inuit Nunangat as a whole, as we state in the intro. As such our investigation of regional trends has to be quite brief given the other aspects we investigate (differences by user, trail type, etc.). We have made two changes which we believe help capture the breadth of the findings. First, figure 4 has been updated to more clearly communicate the pan-Nunangat findings. Further, we have created an interactive portal (https://climatechoices.shinyapps.io/ArcticTrack/) for readers to examine high-level as well as regional findings from the models. We believe that the article now has a very balanced approach to presenting both pan-regional findings as well as describing regional variations.
That said, there are good insights such as reinforcing the fact that trail types are not	Correct. In response we have added text to this paragraph which adds detail on trail

easily substituted for one another. That has pretty much always been the case with sea ice, open water, and land for many reasons, though perhaps the argument here is that this will be even more true in the future?	substitutability and why it may become more challenging in the future.
In summary, the project is a good one in principle, as is the collaborative approach. I would like to see the authors clarify how this article connects to, and builds on, previous work where key principles around thresholds and trail types were developed. Having now read that article, this submitted article becomes much more clear. That being the case, I can see how certain findings are highlighted, but I would like to see a more systematic presentation of results, a fuller discussion of climate variability, some discussion of extreme weather events, and some further probing of the intersectional issues in the discussion of adaptation strategies.	Thank you. We have made the requested changes, as noted above.
Finally some analysis of the reliability of the projections would strengthen the article to justify rounding percentage change in trail access to the nearest one per cent, which looks over-optimistic.	We agree that the variation between climate models is an important back story which has implications for decision makers. As detailed above, we have created an online portal that allows users to examine the results from each GCM that was used in this study. We do feel, however, that additional exploration of climate model precision is beyond the scope of this paper and revolves around a different set of research questions.

Best wishes

James and co-authors

25th Nov 22

Dear Professor Ford,

Your manuscript titled "Projected changes to trail access in the Arctic" has now been seen by our reviewers, whose comments appear below. In light of their advice I am delighted to say that we are happy, in principle, to publish a suitably revised version in Communications Earth & Environment under the open access CC BY license (Creative Commons Attribution v4.0 International License).

We therefore invite you to revise your paper one last time to comply with our format requirements and to maximise the accessibility and therefore the impact of your work. We also attach an annotated manuscript with some additional comments from the editor.

EDITORIAL REQUESTS:

*****Please take care to match our formatting and policy requirements. We will check revised manuscript and return manuscripts that do not comply. Such requests will lead to delays. *****

SUBMISSION INFORMATION:

OPEN ACCESS:

Communications Earth & Environment is a fully open access journal. Articles are made freely accessible on publication under a [CC BY license](http://creativecommons.org/licenses/by/4.0) (Creative Commons Attribution 4.0 International License). This license allows maximum dissemination and re-use of open access materials and is preferred by many research funding bodies.

For further information about article processing charges, open access funding, and advice and support from Nature Research, please visit <https://www.nature.com/commsenv/article-processing-charges>

At acceptance, you will be provided with instructions for completing this CC BY license on behalf of all authors. This grants us the necessary permissions to publish your paper. Additionally, you will be

asked to declare that all required third party permissions have been obtained, and to provide billing information in order to pay the article-processing charge (APC).

[link redacted]

Best regards,

Shin Sugiyama, PhD
Editorial Board Member
Communications Earth & Environment

Clare Davis, PhD
Senior Editor
Communications Earth & Environment

www.nature.com/commsenv/
@CommsEarth

REVIEWERS' COMMENTS:

Reviewer #1 (Remarks to the Author):

I appreciate the authors' clear replies to my earlier comments and the revisions they've made to the manuscript. I enthusiastically recommend that the manuscript be accepted, and I look forward to being able to cite it.

Reviewer #2 (Remarks to the Author):

Great - I think the suggestions are accounted for nicely and I recommend for acceptance/publication.

Reviewer #3 (Remarks to the Author):

The authors have addressed the feedback and criticisms made in the original submission roughly as follows:

1. The authors have done a lot to clarify the methodology, the logic, and links to a previously published paper by reworking the supplementary/methods materials section of the paper. I was pleased to see this and found the section much more digestible and informative. To my mind, readers of the published paper will appreciate the paper much more as a result of this.

2. The use of average/low/high risk categories was problematic while being essential to the methods, argument, and structure of the paper. The authors have helpfully clarified the work that these categories are pragmatic and heuristic. I am persuaded that they are useful and meaningful and lead to a really informative set of conclusions. However I am not entirely persuaded that the specific response to the reviewers that "our work involves some degree of simplification and generalisation" adequately captures what is at stake. In the manuscript, the authors describe these as "archetypes", a term that can be understood in different ways, usually implying some essential or universal form or character. Readers may just accept the use of the term in this context as a metaphor, but it is probably not the right one for describing a combination, cluster or constellation of cultural features in order to construct a category for modelling. Perhaps "heuristic type" might better indicate the idea? Now that the method and types are better outlined in the supplementary materials, the use of 'archetype' may matter less, but the authors may want to consider whether it is serving their purposes.

3. The response to the point about inter-annual variability is good.

4. Figure 4, now updated, is better and the article communicates pan-Nunangat findings better.

5. The most important point is that the revisions communicate the methods and significant findings much better, as well as the key underlying assumptions. This also positions the paper better in terms of future studies that might either develop the paper with further research, or engage with the paper through other kinds of critical analysis.